# Theoretical Modeling Method for Material Removal Characteristics of Abrasive Water Jet Polishing under Rotating Oblique Incidence

**DOI:** 10.3390/mi13101690

**Published:** 2022-10-08

**Authors:** Zhiqiang Zhang, Ci Song, Feng Shi, Guipeng Tie, Wanli Zhang, Bo Wang, Ye Tian, Zhanqiang Hou

**Affiliations:** 1College of Intelligence Science and Technology, National University of Defense Technology, 109 Deya Road, Changsha 410073, China; 2Hunan Key Laboratory of Ultra-Precision Machining Technology, Changsha 410073, China; 3Laboratory of Science and Technology on Integrated Logistics Support, National University of Defense Technology, 109 Deya Road, Changsha 410073, China

**Keywords:** abrasive water jet polishing, material removal characteristics, rotating oblique incidence, theoretical model

## Abstract

Abrasive water jet polishing (AWJP), as an ultra-precision machining technology, has unique machining advantages. However, the machining application of nozzles in vertical and inclined states is greatly limited because rotational symmetric material removal characteristics and the largest amount of central material removal cannot be obtained. At the same time, considering the many controllable and uncontrollable factors in AWJP, it is difficult to accurately model the removal characteristics obtained by machining. Based on the idea of the Preston equation and the calculation of fluid dynamics, this study first analyzed the material removal characteristics of a single abrasive particle and used FLUENT fluid simulation software to obtain the pressure and velocity distributions at different positions in the processes of nozzle rotation and tilt polishing. By analyzing the influence of the pressure and velocity distributions on material removal and the surface shear stress of the workpiece, a theoretical model of the material removal characteristics of abrasive water jet polishing under rotating oblique incidence was established. Finally, the effectiveness of the theoretical removal model was verified by comparing and analyzing experimental and theoretical results.

## 1. Introduction

With the continuous development of modern optical systems, the application range of optical mirrors, especially complex optical surfaces, is becoming more and more extensive and the manufacturing of these optical surfaces with ultra-high accuracy and surface roughness has extremely strict requirements for the processing technology [1]. In order to improve the quality and efficiency of optical parts processing, various new polishing methods have emerged in recent years, such as magnetorheological finishing (MRF) [2], ion beam figuring (IBF) [3], elastic emission machining (EEM) [4], fluid jet polishing (FJP) [5], etc. Among them, abrasive water jet polishing (AWJP) technology mainly realizes material removal through the collision and shear between abrasive particles and workpiece surfaces [6]. It has the ability to modify shape while polishing and can meet the machining requirements of high shape accuracy and high surface roughness that are required for nonlinear and complex curved surface parts. At the same time, it has the characteristics of no thermal influence, no thermal deformation, wide machining range and high machining accuracy [7], which is why it has become one of the current research hot spots.

However, despite the advantages of AWJP, there are still many problems to be solved. In the process of traditional vertical incident jet polishing, a “W” type material removal profile shape with little central material removal is generated on workpiece surfaces, which increases the number of high-frequency errors on the workpiece surfaces. The material removal characteristics are also known as the tool influence function (TIF) [8,9] and are assessed in terms of width, maximum depth and material removal rate [10]. Under the condition of oblique jet incidence, although the amount of material removal in the center of the TIF can be higher than that at the edges, the TIF is “meniscus” shaped and does not have strict symmetry, which brings new difficulties to the calculation of removal material and the allocation of residence time [11]. In order to obtain the ideal Gaussian TIF contour curve, Horiuchi et al. [12] proposed setting several processing points with fixed spacing along the polishing scanning path so that the eccentric rotating nozzle stayed at each processing point for a certain time for processing and finally obtained an axisymmetric “V-shaped” TIF contour. Fang et al. [13] designed a multi-position synthetic impact machining method for vertical impact using a single nozzle that stayed at different positions for a certain amount of time to process and finally obtained the contour shape of the TIF with the largest amount of central material removal. Wang et al. [14] used a magnetic jet device with eccentric rotation motion to study the model removal features under different eccentric distances and obtained the optimal eccentric distance. In this model, the division distribution was closest to the Gaussian features. Peng et al. [15] proposed a jet removal model in which the slit jet rotated around the center and found that the smaller the nozzle diameter, the more concentrated the jet beam energy and that a quasi-Gaussian TIF could be obtained under the condition of vertical jet incidence. Li [16] found that when nozzles were tilted and impinged uniformly around the stagnation points of jet beams, symmetric TIF with the largest amount of central material removal could be obtained. Li et al. [17] also studied the relationship between the position of the incident point of a jet beam on a workpiece surface and material removal morphology under oblique fixed-point incidence by using the polishing method of nozzle tilt rotation and obtained the height range of the nozzle that was required to obtain the corresponding Gaussian TIF. Wang et al. [18] found that when the rotation center of a nozzle coincided with the deepest point of the TIF obtained under the condition of oblique incidence at a fixed point, the obtained TIF was closest to the ideal Gaussian type. 

In terms of theoretical TIF modeling, Cao [10] and Zhou [19] established comprehensive erosion models based on computational fluid dynamics, which cover various process parameters in FJP and can effectively predict material removal characteristics. However, their models are mainly applicable to the vertical jet incidence state. Wang et al. [20] established a universal three-dimensional numerical model of fluid jet polishing based on the CFD method that can be used for vertical jet polishing or oblique jet polishing, but this model cannot represent nozzles that are processed under rotating conditions. Other scholars [11,15,18] have obtained theoretical models of TIF by fitting the mathematical formulae of the material removal profiles obtained by machining. However, models obtained in this way cannot effectively predict material removal during the polishing process. In general, during the process of AWJP, changes in the process parameters, such as slurry concentration, particle size, particle type, slurry pressure, nozzle diameter, standoff distance, jet angle, machining time and the rotating speed of the nozzle, lead to changes in the TIF [21]. Therefore, this study aimed to establish a theoretical model for the material removal characteristics of abrasive water jet polishing under the condition of rotating oblique incidence considering the material removal mechanism of a single abrasive particle so as to effectively predict material removal characteristics during the process of AWJP and better guide polishing experiments.

## 2. Theoretical TIF Modeling

In 1927, Preston put forward the famous Preston hypothesis: for a certain size of polishing die, the rate of polishing removal can be described as the following linear equation in terms of macroscopic effect and a large numerical range [22]:(1)$$\frac{dz}{dt}=Kpv$$
where *K* is the proportionality constant (which is determined by all factors except velocity and pressure), *v* is the relative motion velocity between the instantaneous grinding head and the workpiece at a certain point and *p* is the relative pressure between the grinding head and the workpiece.

Inspired by the basic idea of the Preston equation, TIF during the abrasive water jet polishing process under the condition of rotating oblique incidence was theoretically modeled and analyzed. In the process of AWJP, abrasive particles move with the fluid and their speed gradually increases under its drive, so the abrasive particles have a certain initial speed before colliding with the workpiece. At the same time, the abrasive particles are be subjected to hydrodynamic pressure in the normal direction. During this process, the hydrodynamic pressure determines the maximum indentation depth that can be caused by the abrasive particles during the collision while the horizontal velocity of the abrasive particles along the tangential direction determines the length of the scratches caused by the collision. Therefore, the removal of workpiece material during AWJP is also related to the pressure and velocity. However, the removal rate of workpiece material can only be calculated from the above two parameters and the specific shape of the TIF cannot be obtained; therefore, surface shear stress needs to be introduced [16]. In fact, the fluid carrying the abrasive particles does not leave the surface of the workpiece immediately after collision but continues to move along the surface of the workpiece for a certain distance. During this movement, not only is material removed but surface shear stress is also generated on the surface of the workpiece. At the same time, the distribution of surface shear stress is closely related to the removal of material and the shape of the TIF. Therefore, the theoretical TIF modeling process in this paper was mainly based on the amount of material removal caused by the impact of a single abrasive particle and the influence of the surface shear stress distribution caused by the fluid movement along the workpiece surface.

### 2.1. Calculation of Material Removal Rate in a Single Abrasive Collision Process

In the process of our theoretical analysis, the following hypotheses were put forward:
(1)Assume that the abrasive shape is spherical, its hardness is greater than the hardness of the workpiece and the deformation of the abrasive shape after the collision with the workpiece is negligible;(2)Assume that the abrasive particles are not embedded into the workpiece surface after collision but leave the workpiece at a certain speed;(3)Assume that the slurry medium is uniform;(4)Assume that there is no energy exchange between the abrasive particles.

#### 2.1.1. Calculation of Normal Material Removal Rate

##### Maximum Indentation Depth Caused by a Single Abrasive Particle during Collision

To calculate the material removal rate caused by a single abrasive particle during a single collision, we needed to obtain the maximum indentation depth and the maximum horizontal displacement of the abrasive particle when it leaves the lowest point of the indentation in the workpiece surface. According to the Hertz elastic contact theory [23], when an elastic sphere with radius R is pressed against the surface of a semi-infinite body with load $F_{y}$, the invasion depth is (see Figure 1):(2)$$\lambda_{\max}={\left[\frac{9}{16}\frac{F_{y}{}^{2}}{R}(\frac{1-\upsilon_{1}{}^{2}}{E_{1}}+\frac{1-\upsilon_{2}{}^{2}}{E_{2}})\right]}^{1/3}$$
where $E_{1},\upsilon_{1}$ and $E_{2},\upsilon_{2}$ are the elastic modulus and Poisson’s ratio of the elastic sphere and the semi-infinite body, respectively, R is the radius of the particle and $F_{y}$ is the load acting on the single abrasive particle in the vertical direction.

Figure 2 shows a schematic structural diagram of an abrasive water jet impinging on a workpiece surface under oblique incidence, where X−Y is the nozzle coordinate system with the nozzle center as the origin, $X_{1}-Y_{1}$ is the workpiece surface coordinate system with the stagnation point of the jet as the origin and the jet is sprayed onto the workpiece at a certain speed and inclination angle. In addition, jets can be divided into three parts: free jet zone, surface jet zone and impact zone. In the impact zone, a pressure field is formed due to the collision between the jet fluid and the workpiece. When abrasive water jets are used for polishing at a certain pressure under the oblique incidence condition, the pressure distribution near the surface of the workpiece is similar and can be expressed in the form of normal distribution, whether it is a vertical jet (the jet angle is 90°) or an inclined jet (the jet angle is less than 90°). The pressure distribution at the surface of a workpiece in the impact zone is [24]:(3)$$\frac{P}{P_{m}}=\exp[-0.693{\left(\frac{X_{1}}{b_{p}}\right)}^{2}]$$
where $b_{p}$ is the pressure half-width value, (i.e., the value of $X_{1}$ when $P=P_{m}/2$), *P* represents the pressure distribution at the workpiece surface in the impact zone and $P_{m}$ is the maximum pressure at the stagnation point *S* (see Figure 2). In addition, due to the impact of jet fluid on a workpiece in the impact zone, the kinetic energy of the abrasive particles is gradually reduced and converted into pressure wave energy. Therefore, the pressure at the stagnation point can be expressed as [25]:(4)$$P_{m}=\frac{1}{2}\rho {\left(u_{m0}\sin\alpha \right)}^{2}$$
where ρ is the density of the abrasive particles, $u_{m0}$ is the axis velocity at the junction between the free jet zone and impact zone and α is the angle between the jet beam and the plane of the workpiece (i.e., the jet angle). On the other hand, the distance between the stagnation point and the intersection between the central axis of the jet beam and the workpiece surface, which is indicated by *O* (see Figure 2), can be approximately expressed as:(5)$$\frac{e}{L}=0.154\cot\alpha$$
where e represents the eccentricity of the stagnation point and L represents the distance from point *O* to the nozzle outlet.

At the same time, since changes in any parameter in the machining process lead to changes in the shape of the TIF and since it is impossible to determine the position of the maximum material removal point on the $X_{1}$ axis in a theoretical analysis, even when the position of the stagnation point and the jet incidence point are known, our theoretical modeling analysis was conducted assuming that the nozzle rotated around the axis where the stagnation point was located (i.e., the $Y_{1}$ axis). Since TIF obtained by rotating oblique incidence machining has symmetry [16], this study focused on analyzing and calculating the material removal characteristics along the positive direction of the $X_{1}$ axis. In addition, due to the limitations of the experimental conditions, the pressure and velocity on the workpiece surface at any time during rotary oblique incidence processing could not be directly obtained. Therefore, we used FLUENT software to simulate AWJP under the condition of oblique incidence and studied the situation of nozzle rotating machining by analyzing the change laws of pressure and velocity at various points in the impact zone during the process. 

Although jets act on the entire impact zone, when the nozzles rotate around point S, there are certain law and periodic changes in the pressure and velocity at each position on the $X_{1}$ axis and the minimum period is the time required for the nozzles to rotate through one cycle. In order to explore this rule, we used FLUENT software to establish a three-dimensional model of the jet spraying process (as shown in Figure 3), where the nozzle diameter was 1 mm, the jet angle was 60°, the standoff distance was 13 mm, the inlet was the velocity inlet, the initial velocity was set as 28 m/s, the outlet was the pressure outlet and the jet was sprayed onto the workpiece from the top right. Figure 4 shows a pressure cloud diagram of the workpiece, from which it can be clearly observed that the maximum point of pressure on the workpiece appeared directly behind point O and that the pressure was symmetrically distributed about the $X_{1}$ axis. 

As shown in Figure 4, we took a straight line that coincided with the $X_{1}$ axis and numbered it as 1 and then took a straight line passing through the point *S* every 15° in the clockwise direction and numbered it as 2~12 (except the straight line where the $Y_{1}$ axis was located). Then, when the nozzle rotated and polished around the point S, a straight line coincided with the $X_{1}$ axis every 15°. At this time, the points on the straight lines that were the same distance to the point *S* coincided with the points on the $X_{1}$ axis in turn. Meanwhile, we assumed that the slurry medium was uniform, so the abrasive particles in any cross-section that passed through the point S in the jet beam were uniformly arranged and their number did not change. In Figure 5a,b are the pressure curves on the straight line numbered 1~12 and (c) and (d) are the velocity curves. Since the fluid velocity on the workpiece surface was 0, the velocity 5 μm away from the workpiece surface was taken. In the polishing process, as the nozzle rotated around the point S at a certain angular velocity, the points with the same radius as the center of the point S on the straight line numbered 1~12 overlapped the $X_{1}$ axis in turn. We took the values of the points on the straight lines at distances of 0.8 mm, 1.0 mm, 1.2 mm, 1.4 mm and 1.6 mm from the point S and normalized them to obtain the scatter diagram shown in Figure 6. It can be seen that the pressure decreased gradually with the increase in the distance from the point S and that the pressure was at its maximum at the stagnation point. The velocity increased first and then decreased with the increase in the distance from the point S. In addition, the pressure and velocity at certain points on the $X_{1}$ axis changed with the rotation of the nozzle in a certain trend. In order to understand this change rule, we took the average values of the pressure and velocity at each position and used MATLAB to fit the values to obtain the red curves in Figure 6a,b. The expressions of these two curves were:(6)$$\begin{array}{l} P(t)=0.3121+0.4569\cos(\omega t)-9.985\times 10^{-18}\sin(\omega t)+0.2134\cos(2\omega t)+1.677\times 10^{-17}\sin(2\omega t)+0.06981\cos(3\omega t)\\ \;\;+1.184\times 10^{-17}\sin(3\omega t)-0.006116\cos(4\omega t)-1.287\times 10^{-17}\sin(4\omega t)-0.01749\cos(5\omega t)-3.687\times 10^{-17}\sin(5\omega t)\\ \;\;-0.02316\cos(6\omega t)+4.207\times 10^{-18}\sin(6\omega t)-0.01604\cos(7\omega t)+5.388\times 10^{-17}\sin(7\omega t) \end{array}$$
(7)$$\begin{array}{l} V(t)=0.6453+0.3847\cos(\omega t)-4.962\times 10^{-18}\sin(\omega t)-0.0452\cos(2\omega t)-2.896\times 10^{-17}\sin(2\omega t)+0.08532\cos(3\omega t)\\ \;\;-3.467\times 10^{-17}\sin(3\omega t)-0.0797\cos(4\omega t)-4.153\times 10^{-17}\sin(4\omega t)+0.01038\cos(5\omega t)-3.272\times 10^{-17}\sin(5\omega t)\\ \;\;-0.006912\cos(6\omega t)-3.528\times 10^{-17}\sin(6\omega t)+0.01288\cos(7\omega t)-3.519\times 10^{-17}\sin(7\omega t) \end{array}$$
where P(t) represents the normalized pressure value, V(t) represents the normalized velocity value, ω is the angular velocity of the nozzle rotating around the stagnation point, ω=2πn, n is the rotation speed of the nozzle and t is the rotation time of the nozzle. Then, the pressure at each point in the positive direction of the $X_{1}$ axis at different times could be expressed as:(8)$$P(X_{1},t)=p_{m}p(t)\exp[-0.693{\left(\frac{X_{1}}{b_{p}}\right)}^{2}]$$

If the time and distance in the positive direction along the $X_{1}$ axis were discretized, the pressure at each position at different times could be expressed in a matrix, as follows:(9)$${\left[P(X_{1},t)\right]}_{n_{1}\times n_{2}}={\left[P\right]}^{T}{}_{n_{1}\times 1}\times {\left[P(t)\right]}_{1\times n_{2}}=p_{m}\cdot {\left[\frac{P}{P_{m}}\right]}^{T}{}_{n_{1}\times 1}\times {\left[P(t)\right]}_{1\times n_{2}}$$
where $n_{1}$ and $n_{2}$ represent the number of discrete parts in the distance and time, respectively. Equation (9) could be used to predict the pressure at different times and positions on the $X_{1}$ axis during the polishing process when the nozzle was at a certain angular velocity under the rotating oblique incidence condition. 

Since the velocity of fluid along the axis of a nozzle decreases after it is ejected from the nozzle, the pressure at the outlet of the nozzle is inconsistent with the pressure on the surface of the workpiece. As shown in Figure 2, fluid is less affected by the surface of a workpiece in the free jet zone, so the change in velocity is also small. However, when the fluid reaches the impact zone, the velocity along the axis of the nozzle decays rapidly and after the fluid reaches the workpiece surface, the jet velocity gradually increases along the workpiece surface to both sides. According to [26], the actual effective impact velocity is the instantaneous velocity at the junction between the free jet zone and the impact zone when Y=0.86H and its magnitude can be expressed as follows:(10)$$u_{m0}=\frac{2.4u_{0}}{\sqrt{\frac{0.86H}{d\sin\alpha }-2.5}}$$
where H is the standoff distance, $u_{m0}$ is the actual effective impact velocity and $u_{0}$ is the velocity of the fluid at the nozzle outlet. When the jet flow is Q, the velocity of the fluid at the nozzle outlet can be expressed as:(11)$$u_{0}=\frac{4Q}{\pi d^{2}}$$
where d is the nozzle diameter. 

In combination with Equations (4) and (9)–(11), the load received by a single abrasive particle at different positions and at different times on the $X_{1}$ axis could be expressed as:(12)$${\left[F_{y}(X_{1},t)\right]}_{n_{1}\times n_{2}}=[P(X_{1},t)]\cdot \pi R^{2}=\frac{46.08\rho R^{2}Q^{2}\sin^{2}\alpha }{(\frac{0.86H}{d\sin\alpha }-2.5)\pi d^{4}}\cdot {\left[\frac{P}{P_{m}}\right]}^{T}{}_{n_{1}\times 1}\times {\left[p(t)\right]}_{1\times n_{2}}$$

Therefore, after a single abrasive particle was sprayed to the workpiece, the maximum indentation depth at any time $\left(t=t_{1}\right)$ and at any position $\left(X_{1}=x_{1}\right)$ could be expressed as:(13)$$\lambda_{\max}(x_{1},t_{1})={\left[\frac{9}{16}\frac{\left[F_{y}{\left(x_{1},t_{1}\right)}^{2}{}_{1\times 1}\right]}{R}{\left(\frac{1-\upsilon_{1}{}^{2}}{E_{1}}+\frac{1-\upsilon_{2}{}^{2}}{E_{2}}\right)}^{2}\right]}^{1/3}$$

##### Material Removal Rate in the Normal Direction for a Single Abrasive Collision

Figure 7 shows a schematic simulation diagram of the fluid streamline movement trajectory of AWJP under oblique incidence. According to [27], in the process of AWJP, when the abrasive particle size is greater than 10 μm, the particles directly collide with the workpiece surface along the initial incidence direction. When the particle size is less than 100 nm, the abrasive particle trajectory basically coincides with the movement trajectory of the jet stream. The modeling process in this paper assumed that the abrasive particle size was between 100 nm and 10 μm and that the abrasive particles in the slurry would first impact and collide with the workpiece along the incident direction and then move along the streamline track on the workpiece surface under the action of the fluid.

As can be seen from the above analysis, the velocity of the jet fluid at the junction between the impact zone and the free jet zone is $u_{m0}$ and the velocity decays rapidly after the fluid enters the impact zone. Therefore, we assumed that the velocity of the abrasive particles before the collision with the workpiece was $u_{m0}$ (see Figure 8). In addition, the velocity before collisions between abrasive particles and workpieces is large and the displacement caused by the collisions is relatively small, so the collision time is extremely short. When abrasive particles collide with workpieces, the velocity along the normal direction decreases sharply and finally becomes zero. According to the law of momentum conservation, the resistance of workpiece surfaces acting on abrasive particles at the moments when the abrasive particles collide with the workpieces is much larger than that of other forms of acting forces [28], which can be expressed as:(14)$$F_{t}\sin\alpha =\frac{m\Delta u_{mn}}{\Delta t}$$
where $\Delta u_{mn}$ is the change in the velocity component in the normal direction, $\Delta u_{mn}=u_{m0}\sin\alpha$, Δt is the time taken for the abrasive particles to reach the lowest point of the indentation and $F_{t}$ is the resistance of the workpiece to the abrasive particles during collision. Therefore, when analyzing the process of collisions between abrasive particles and workpieces, other forces can be ignored, i.e., the velocity of the abrasive particles in the normal direction decreases sharply. When the abrasive particles reach the lowest point of the indentation, the velocity in the normal direction becomes 0. Since the abrasive particles also have a certain initial velocity in the horizontal direction, they also move a certain distance in the horizontal direction when they reach the lowest point of the indentation along the normal direction. However, this distance is extremely small relative to the diameter of the abrasive particles. Therefore, when calculating the material removal amount caused by abrasive particles in the normal direction during the initial collision, the displacement of the abrasive particles in the horizontal direction can be ignored.

A schematic diagram of the forces before a single abrasive particle collides with a workpiece and reaches the lowest point of indentation is shown in Figure 8. Note that the cross-sectional area of the contact surface in the horizontal direction when the abrasive particle collides with the workpiece is $A_{y}$, then the material removal rate at any time $\left(t=t_{1}\right)$ and any position $\left(X_{1}=x_{1}\right)$ during the process of the collision between the abrasive particles and the workpiece surface to the lowest point of the indentation can be expressed as: (15)$$V_{y}(x_{1},t_{1})=\int_{0}^{\lambda_{\max}(x_{1},t_{1})}A_{y}d\lambda =\int_{0}^{\lambda_{\max}(x_{1},t_{1})}\pi [R^{2}-{\left(R-\lambda \right)}^{2}]d\lambda =\pi (R\lambda_{\max}{}^{2}-\frac{1}{3}\lambda_{\max}{}^{3})|{}_{\left(x_{1},t_{1}\right)}$$
where λ represents the pressing depth of a single abrasive particle at a certain time before reaching the lowest point of indentation and $t_{1}$ represents a certain time during the nozzle rotation. Since the time required for an abrasive particle to collide with a workpiece is extremely short, the time required for a single abrasive particle to reach the lowest point of indentation can be ignored in the calculation process; therefore, λ is independent of $t_{1}$ in the calculation process.

#### 2.1.2. Calculation of Tangential Material Removal Rate

In this study, it was assumed that the hardness of the abrasive particles was greater than that of the workpiece and that the influence of deformations due to collisions could be ignored. According to [29], when abrasive particles vertically impact a workpiece surface at a high speed (10~500 m/s), until the normal speed is 0, about 90% of the initial energy is dissipated in the plastic deformation of the workpiece. Therefore, the influence of the elastic effect of the workpiece can be ignored during the movement of abrasive particles from the lowest point to the surface of the workpiece, i.e., abrasive particles do not rebound after reaching the lowest point of indentation on the surface of a workpiece but continue to move along the direction of the fluid movement under the action of the fluid.

An overall force diagram of abrasive particles from the lowest point of indentation to before leaving a workpiece surface is shown in Figure 9. When slurry carrying abrasive particles is sprayed onto a workpiece surface, due to the impact of the fluid on the workpiece surface, the velocity of the fluid and that of the abrasive particles in the fluid sharply decrease and then gradually increase along the workpiece surface. Moreover, due to the impact and shear action between abrasive particles and workpiece surfaces, a large velocity difference is formed between the two in the fluid movement direction (see Figure 2). The fluid shear velocity in the positive direction of the $X_{1}$ axis in the impact zone can be expressed as [24]:(16)$$\frac{u_{*}}{u_{0}}\sqrt{\frac{L}{d}}=0.17{\left[erf(6.2\frac{X_{1}}{L})\right]}^{1/2}$$
where $u_{*}$ represents the shear velocity of the fluid. Since nozzles rotate around a stagnation point, the shear velocity at different points in the positive direction of the $X_{1}$ axis is different. According to Equation (7), the shear velocity at different points at different times and positions in the positive direction of the $X_{1}$ axis can be expressed as:(17)$${\left[u_{*}(X_{1},t)\right]}_{n_{1}\times n_{2}}={\left[u_{*}\right]}^{T}{}_{n_{1}\times 1}\times {\left[V(t)\right]}_{1\times n_{2}}=(u_{0}\sqrt{\frac{d}{L}})\cdot {\left[\frac{u_{*}}{u_{0}}\sqrt{\frac{L}{d}}\right]}^{T}{}_{n_{1}\times 1}\times {\left[V(t)\right]}_{1\times n_{2}}$$

Therefore, due to the velocity difference between the abrasive particles and the fluid along the direction of fluid movement, the drag force on a single abrasive particle at any time $\left(t=t_{1}\right)$ and at any position $\left(X_{1}=x_{1}\right)$ can be expressed as [30]:(18)$$F_{s}(x_{1},t_{1})=\frac{1}{2}\pi C_{D}(x_{1},t_{1})R^{2}\rho_{L}\left|{\left[u_{*}(x_{1},t_{1})\right]}_{1\times 1}-u_{p}\right|\{{\left[u_{*}(x_{1},t_{1})\right]}_{1\times 1}-u_{p}\}$$
where $F_{s}$ represents the drag force, $u_{p}$ is the abrasive particle velocity, R is the abrasive particle radius, $\rho_{L}$ is the fluid density and $C_{D}(x_{1},t_{1})$ is the Stokes drag coefficient, which can be expressed as follows:(19)CD(x1,t1)=24Re(x1,t1)
where Re is the particle Reynolds number, which can be expressed as:(20)Re(x1,t1)=2RρL|[u∗(x1,t1)]1×1−up|μ
where μ is the viscosity of the fluid. At the same time, there is also a velocity gradient along the normal direction of the motion of the fluid, so abrasive particles are also affected by the Magnus force and Saffman force, the magnitude of which can be expressed as [31]:(21)$$F_{M}(x_{1},t_{1})=\pi R^{3}\rho_{L}\omega^{\prime }\{{\left[u_{*}(x_{1},t_{1})\right]}_{1\times 1}-u_{p}\}$$
(22)$$F_{sm}(x_{1},t_{1})=1.62{\left(2R\right)}^{2}\sqrt{\rho_{L}\mu }\{{\left[u_{*}(x_{1},t_{1})\right]}_{1\times 1}-u_{p}\}\sqrt{\left|\frac{du_{*}}{dY_{1}}\right|}$$
where $F_{M}$ and $F_{sm}$ represent the Magnus force and Saffman force, respectively, $\omega^{\prime }$ is the rotational angular velocity of the abrasive particle itself and $du_{*}/dY_{1}$ is the velocity gradient of the fluid along the $Y_{1}$ direction.

In addition, since the horizontal displacement of abrasive particles during collisions is very small, it always occurs in the impact zone throughout the whole collision process, so it is also affected by the pressure of the jet and the supporting force of the workpiece. Meanwhile, abrasive particles also have their own gravity and the impact of the friction force on abrasive particles can be ignored in this process. Therefore, when abrasive particles leave the lowest point of indentation in a workpiece surface, the equation of motion at any time $\left(t=t_{1}\right)$ and at any position $\left(X_{1}=x_{1}\right)$ in the horizontal direction and normal direction can be expressed as:(23)$$m\frac{du_{pX_{1}}}{dt}=F_{s}(x_{1},t_{1})$$
(24)$$m\frac{du_{pY_{1}}}{dt}={\left(F_{sm}+F_{M}+F_{z}-F_{y}-G\right)|}_{\left(x_{1},t_{1}\right)}$$
where $F_{y}$ is the pressure of the jet, $F_{z}$ is the supporting force of the workpiece, G is the gravity of the abrasive particles, m and ρ are the mass and density of the abrasive particles, respectively, and $m=(4/3)\pi R^{3}\rho$. At the same time, since the collision time between abrasive particles and workpieces is extremely short, it is assumed that the magnitude and direction of the force on a single abrasive particle are constant throughout the process of movement, so its acceleration in the horizontal and normal directions does not change. Therefore, for any time $\left(t=t_{1}\right)$ and position $\left(X_{1}=x_{1}\right)$, the motion equation for the displacement of abrasive particles in the horizontal and normal directions with varying motion times can be expressed as follows:(25)$$\lambda (x_{1},t_{1})={\left(\lambda_{\max}-\frac{1}{2}a_{p}{{}_{Y}}_{{}_{1}}{t^{\prime }}^{2}\right)|}_{\left(x_{1},t_{1}\right)}$$
(26)$$l^{\prime }(x_{1},t_{1})={\left(u_{p1}+\frac{1}{2}a_{p}{}_{X_{1}}{t^{\prime }}^{2}\right)|}_{\left(x_{1},t_{1}\right)}$$
where $t^{\prime }$ represents the movement time of the abrasive particles, λ and $l^{\prime }$ represent the displacement of the abrasive particles in the horizontal direction and the normal direction, respectively (both of which are functions of $t^{\prime }$,$a_{X_{1}}=du_{pX_{1}}/dt$ and $a_{Y_{1}}=du_{pY_{1}}/dt$), and $u_{p1}$ represents the velocity of an abrasive particle along the tangential direction when it reaches the lowest point of indentation. The relationship between the displacement of a single abrasive particle in the horizontal direction and its displacement in the normal direction at any time $\left(t=t_{1}\right)$ and any position $\left(X_{1}=x_{1}\right)$ can be obtained as follows:(27)$$l^{\prime }|(x_{1},t_{1})={\left[u_{p1}\sqrt{\frac{2(\lambda_{\max}-\lambda )}{a_{Y_{1}}}}+\frac{a_{X_{1}}}{a_{Y_{1}}}(\lambda_{\max}-\lambda )\right]|}_{\left(x_{1},t_{1}\right)}={f(\lambda )|}_{\left(x_{1},t_{1}\right)}$$

As shown in Figure 9, $A_{x}$ is the cross-sectional area of the contact surface between a single abrasive particle and a workpiece surface in the normal direction when it moves from the lowest point of indentation to the workpiece surface, which can be expressed as:(28)$$\begin{array}{l} {A_{x}|}_{\left(x_{1},t_{1}\right)}={\left[\frac{2\theta \pi R^{2}}{360}-(R-\lambda )\sqrt{R^{2}-{\left(R-\lambda \right)}^{2}}\right]|}_{\left(x_{1},t_{1}\right)}={\left[\frac{\pi R^{2}}{180}\arccos\frac{R-\lambda }{R}-(R-\lambda )\sqrt{R^{2}-{\left(R-\lambda \right)}^{2}}\right]|}_{\left(x_{1},t_{1}\right)}\\ \;\;\;\;={g(\lambda )|}_{\left(x_{1},t_{1}\right)} \end{array}$$

Therefore, the amount of removal material in the horizontal direction caused by a single abrasive particle at any time $\left(t=t_{1}\right)$ and position $\left(X_{1}=x_{1}\right)$ can be expressed as:(29)$${V_{x}|}_{\left(x_{1},t_{1}\right)}={\left(\int_{0}^{{l^{\prime }}_{\max}}A_{x}dl^{\prime }\right)|}_{\left(x_{1},t_{1}\right)}={\left(\int_{0}^{{l^{\prime }}_{\max}}g(\lambda )f^{\prime }(\lambda )d\lambda \right)|}_{\left(x_{1},t_{1}\right)}$$

Combined with Equations (15) and (29), we obtained that the volume of material removed by a single abrasive particle at any time $\left(t=t_{1}\right)$ and position $\left(X_{1}=x_{1}\right)$ in a collision process can be expressed as:(30)$${V_{all}|}_{\left(x_{1},t_{1}\right)}={\left(V_{x}+\frac{1}{2}V_{y}\right)|}_{\left(x_{1},t_{1}\right)}$$

### 2.2. The Number of Abrasive Particles Colliding with a Workpiece Surface per Unit Time and Area

It was pointed out in [32] that the number of abrasive particles colliding with a workpiece surface per unit time can be calculated from the concentration of abrasive particles in the slurry and the flow rate of the jet. At the same time, since fluid is turbulent after ejection from a nozzle, its transverse pulsation continuously exchanges mass and momentum between the fluid and surrounding stationary gas and drives the surrounding gas flow, so the mass flow rate and cross-sectional area of the jet increase along its forward direction [33]. Therefore, the actual cross-sectional area of a jet that is in contact with a workpiece surface is not the same as the outlet area of the nozzle. In addition, considering the influence of nozzle diameter, standoff distance and slurry pressure on the working area of a polishing area, the effective number of abrasive particles per unit area should be taken into account when calculating the material removal rate. Therefore, the number of abrasive particles per unit time and per unit area colliding with a workpiece surface can be expressed as [32]:(31)$$N^{\prime }=\frac{3u_{0}\rho_{L}c}{4\rho \pi k^{2}R^{3}}$$
where c is the concentration of abrasive particles in the slurry and k is the proportionality constant. When the standoff distance is small, its influence on k can be ignored. The value of k is also related to the slurry pressure in that the larger its value, the larger the polishing area. Generally, the value of k is about 2–3.

### 2.3. Distribution of Shear Stress on Workpiece Surfaces

In the process of AWJP, the shear stress generated by the radial flow of abrasive particles in slurry on a workpiece surface is the key to material removal [34]. According to the experimental results of Beltaos et al. [24], shear stress distribution on a workpiece surface can be expressed as follows:(32)$$\frac{\tau_{0}}{\tau_{0m}}=0.18\frac{L}{X_{1}}-(0.18\frac{L}{X_{1}}+9.43\frac{X_{1}}{L})\exp[-114{\left(\frac{X_{1}}{L}\right)}^{2}]$$
where $\tau_{0m}$ is the maximum surface shear stress, L is the linear distance between the incident point of the jet flow from the nozzle and the workpiece surface (see Figure 2) and the surface resistance coefficient can be expressed as follows:(33)Cr=τ0mρLuL12/2=0.0474ReL−4/5
where $u_{L1}$ is the maximum fluid velocity along the positive direction of the $X_{1}$ axis, ReL is the Reynolds number of the fluid (ReL=ρLu0d/η) and η is the dynamic viscosity coefficient of the polishing fluid.

As shown in Figure 2, $u_{L}$ represents the fluid velocity in the impact zone and its maximum value can be expressed as [24]:(34)$$\frac{u_{L1}}{u_{0}}\sqrt{\frac{L}{d}}=2.77{\left\{1-\exp[-38.5{\left(\frac{X_{1}}{L}\right)}^{2}]\right\}}^{1/2}$$

According to Equation (7), the maximum fluid velocity at different times and different positions along the positive direction of the $X_{1}$ axis can be expressed as:(35)$${\left[u_{L1}(X_{1},t)\right]}_{n_{1}\times n_{2}}={\left[u_{L1}\right]}^{T}{}_{n_{1}\times 1}\times {\left[V(t)\right]}_{1\times n_{2}}=(u_{0}\sqrt{\frac{d}{L}})\cdot {\left[\frac{u_{L1}}{u_{0}}\sqrt{\frac{L}{d}}\right]}^{T}{}_{n_{1}\times 1}\times {\left[V(t)\right]}_{1\times n_{2}}$$

According to Equation (33), the maximum surface shear stress can be obtained when $u_{L1}$ is at its maximum value. Since nozzles rotate around a stagnant point, the maximum value of fluid velocity is different at different times along the positive direction of the $X_{1}$ axis, so the maximum values of $u_{L1}$ can be used to form a separate matrix, namely ${\left[u_{L1\max}\right]}_{1\times n_{2}}$. Therefore, the maximum shear stress at different times can be expressed as:(36)[τ0m(t)]1×n2=(0.237ρLReL−4/5)⋅[uL1max]⋅[uL1max]

Therefore, shear stress at different times and different positions along the positive direction of the $X_{1}$ axis can be expressed as:(37)$${\left[\tau_{0}(t,X_{1})\right]}_{n_{2}\times n_{1}}={\left[\tau_{0m}(t)\right]}^{T}{}_{n_{2}\times 1}\times {\left[\frac{\tau_{0}}{\tau_{0m}}\right]}_{1\times n_{1}}$$

In summary, during the polishing process of AWJ under the condition of rotating oblique incidence, the material removal depth at different times and different positions along the positive direction of the $X_{1}$ axis can be expressed as:(38)$${\left[Z(X_{1},t)\right]}_{n_{1}\times n_{2}}={\left[\tau_{0}(t,X_{1})\right]}^{T}{}_{n_{1}\times n_{2}}\cdot {\left[V_{all}(X_{1},t)\right]}_{n_{1}\times n_{2}}\cdot \frac{tN^{\prime }}{n_{2}}k_{1}k_{2}$$
where $k_{1}$ is the unit correction coefficient, $k_{1}={\left(k_{3}G\right)}^{-1}$, $k_{3}$ is generally the number of abrasive particles emitted from the nozzle in a unit time and $k_{2}$ is the numerical correction coefficient.

## 3. Experimental Verification and Discussion

Figure 10a shows an abrasive water jet machining device under oblique incidence, which can rotate uniformly around the C-axis and the angle between the nozzle and the workpiece surface can be adjusted in the range of 0°~90°. At the same time, it can also transversally translate to adjust the distance between the incident point of the jet on the workpiece surface and the center of the rotation around the C-axis. In this experiment, CeO_2_ slurry was used to process K9 glass with the following parameters: the slurry concentration was 3.5%, the average particle size was 3 μm, the jet angle was 60°, the standoff distance was 13 mm, the nozzle diameter was 1 mm, the device rotation speed was 1 r/min, the slurry pressure was 1.5 MPa and the machining time was 5 min. Figure 10b shows the TIF curve obtained under the condition of stationary oblique incidence machining. In the process of rotational machining, since the positional relationship between the maximum point of material removal and the center of the rotational axis could not be determined, the rotation center of the nozzle could be located between a and b or b and c but the shape of the TIF obtained by polishing around the different centers of rotation was different.

The experimental results are shown in Figure 11: the black curve shown in Figure 11a is the experimental result obtained when the rotation center of the nozzle was located between point a and point b in Figure 10b; the black curve in Figure 11b is the experimental result obtained when the rotation center was located at point b in Figure 10b; the red curves in Figure 11a,b are the theoretical calculation curves under the two processing conditions.

It can be seen in Figure 11a that when the rotation center of the nozzle was located between points a and b in Figure 10b, the TIF curve obtained by processing had two peak points, which was not an ideal Gaussian-like TIF curve with the largest amount of central material removal. At the same time, the TIF curve (i.e., the red curve) calculated theoretically using Equation (38) matched the experimental curve well. In addition, by comparing the theoretical TIF curves, it was also found that the distance between the rotation center of the nozzle and the stagnation point was about 0.3 mm and that the distance from the peak point was about 0.8 mm. According to Equation (5), under the conditions of H=13mm and α=60°, the eccentricity of the stagnation point could be calculated to be e=1.335mm. So, the distance from the stagnation point to the peak point was less than the distance from the stagnation point to the jet incidence point, which was consistent with the theoretical results.

According to the theoretical calculation results, the nozzle was moved 0.8 mm away from the C axis and processed under the same experimental parameters to obtain the black TIF curve in Figure 11b. It was found that the TIF curve at this time only had one peak point and the center was symmetric, so the maximum amount of material removal changed slightly compared to that before adjustment. Therefore, it was reasonable to conclude that the rotation center of the nozzle passed through the maximum point of material removal at this time. In addition, the red curve obtained using the theoretical calculations also matched the experimental curve well, which proved that the theoretical TIF model could predict the contours of the TIF and the maximum material removal depth obtained by the nozzle under the condition of rotating oblique incidence well. It is worth noting that there were certain deviations between the theoretical TIF curve and the experimental TIF curve, which might have been caused by fluctuations in slurry pressure, the instability of the slurry concentration and abrasive uniformity or the tilt of the workpiece surface, among other options.

## 4. Conclusions

In this paper, based on the idea of the Preston equation and the calculation of fluid dynamics and starting from the material removal characteristics of a single abrasive particle combined with the use of FLUENT simulation software to analyze the distributions of pressure and shear stress on a workpiece surface, a theoretical model was established for the material removal characteristics of AWJP under the condition of rotating oblique incidence. The effectiveness of the model was verified by comparing TIF curves that were obtained experimentally and theoretically. The main conclusions were as follows:(1)By using FLUENT simulation software to analyze the velocity and pressure distributions on a workpiece surface when a nozzle was at rest and under oblique incidence processing, the variation rule for time when the nozzle rotated at a certain angular velocity could be obtained, according to the symmetry of the distribution. At the same time, by discretizing the time and distance, the material removal depth at any position and any time in the impact zone could also be quantitatively analyzed.(2)By comparing and analyzing experimental TIF curves and theoretical TIF curves, it was verified that the model could effectively predict the contours of the TIF and the maximum material removal depth obtained when the nozzle was under the condition of rotating oblique incidence. It is worth noting that there were certain deviations between the theoretical TIF curves and the experimental TIF curves, which might have been caused by fluctuations in slurry pressure, the instability of the slurry concentration and abrasive uniformity or the tilt of the workpiece surface, among other options.

## Figures and Tables

**Figure 1 micromachines-13-01690-f001:**
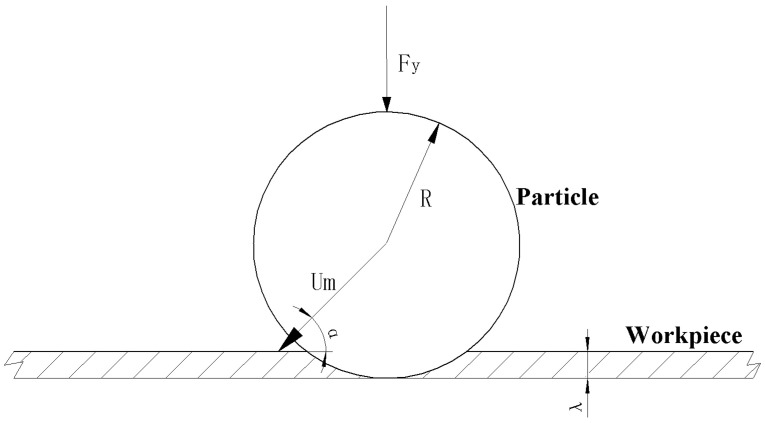
A schematic diagram of the erosion from a single abrasive particle.

**Figure 2 micromachines-13-01690-f002:**
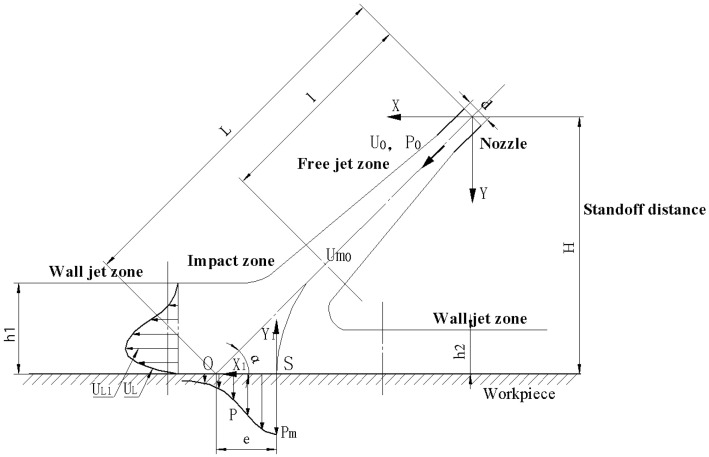
A schematic diagram of the oblique incidence structure of AWJ.

**Figure 3 micromachines-13-01690-f003:**
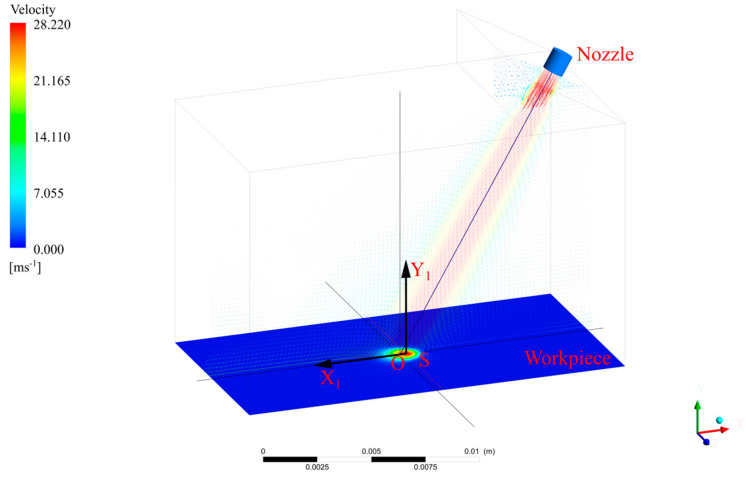
A schematic diagram of the FLUENT jet simulation structure.

**Figure 4 micromachines-13-01690-f004:**
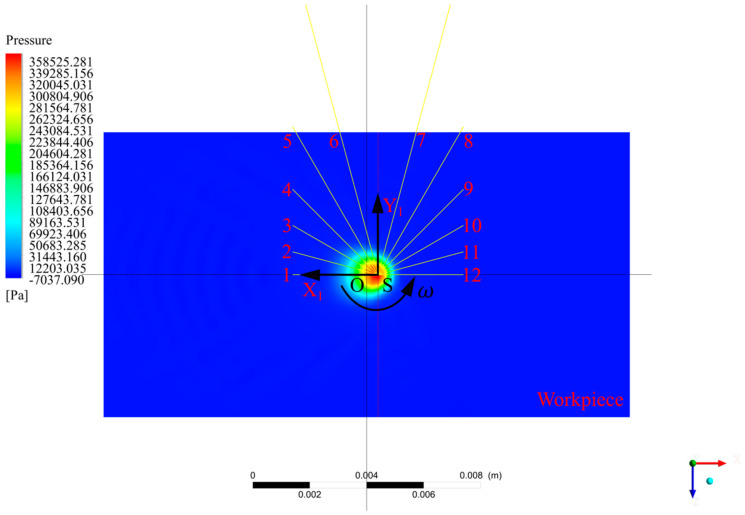
A pressure cloud diagram of a workpiece surface.

**Figure 5 micromachines-13-01690-f005:**
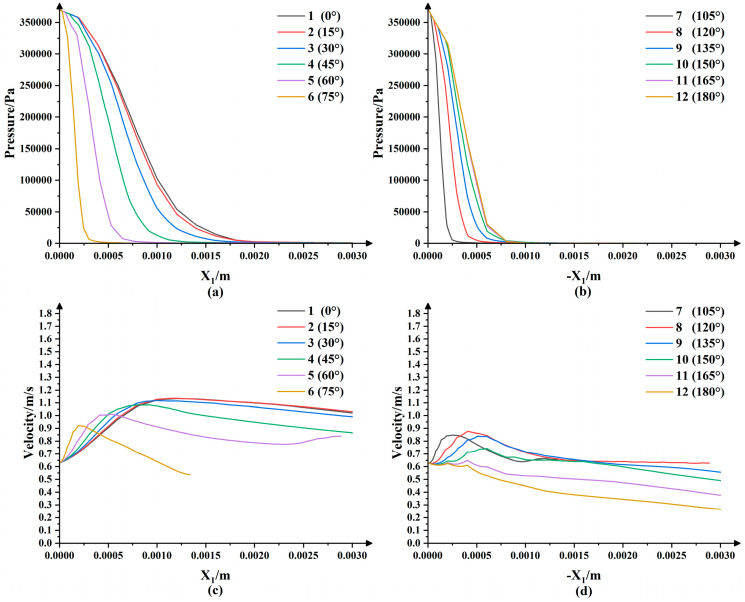
Simulation curves of pressure and velocity at different angles on the workpiece surface: (**a**) pressure curves at 0~75°; (**b**) pressure curves at 105~180°; (**c**) velocity curves at 0~75°; (**d**) velocity curves at 105~180°.

**Figure 6 micromachines-13-01690-f006:**
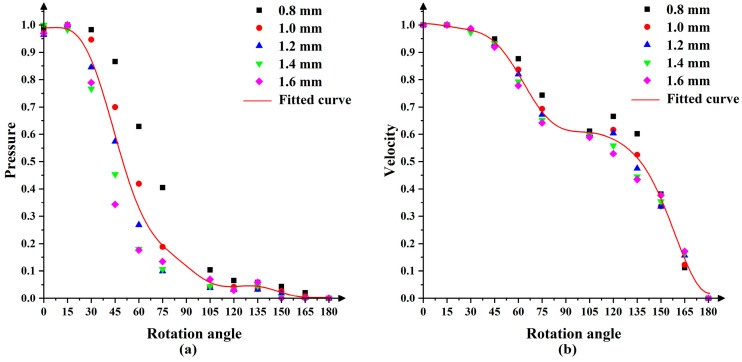
Normalized scatter plots of (**a**) pressure and (**b**) velocity with their fitted curves.

**Figure 7 micromachines-13-01690-f007:**
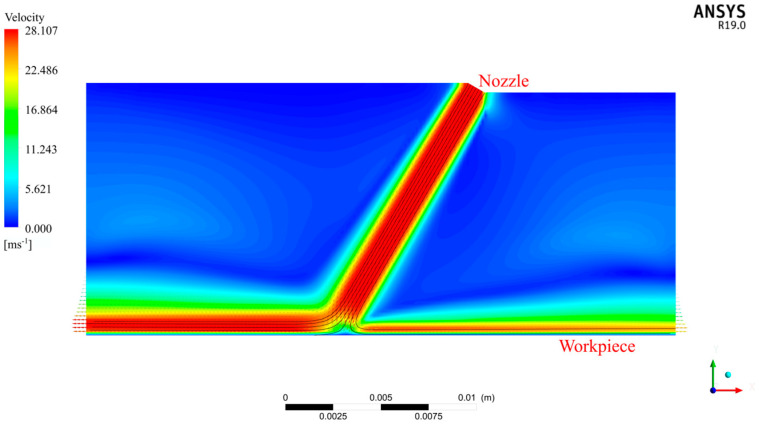
A simulation diagram of the streamline trajectory of jet polishing.

**Figure 8 micromachines-13-01690-f008:**
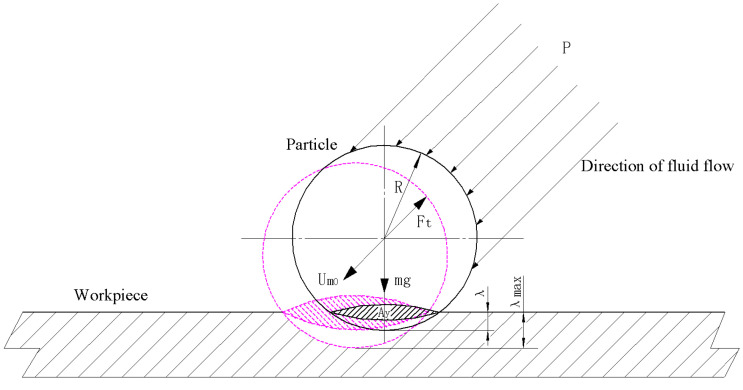
A motion diagram of an abrasive particle before reaching the lowest point of indentation.

**Figure 9 micromachines-13-01690-f009:**
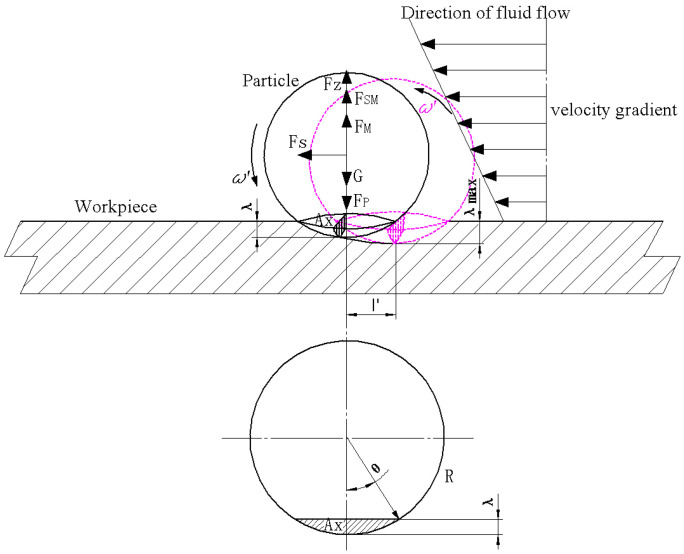
A schematic diagram of the force of abrasive particles during the process from the lowest point of indentation to leaving the workpiece surface.

**Figure 10 micromachines-13-01690-f010:**
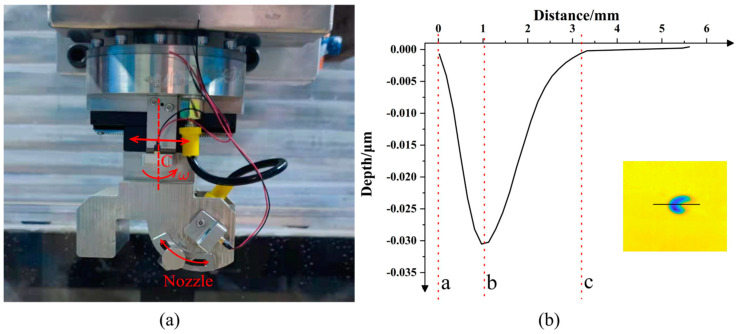
(**a**) An AWJP device; (**b**) the section curve of the TIF obtained when the nozzle was tilted at 60°.

**Figure 11 micromachines-13-01690-f011:**
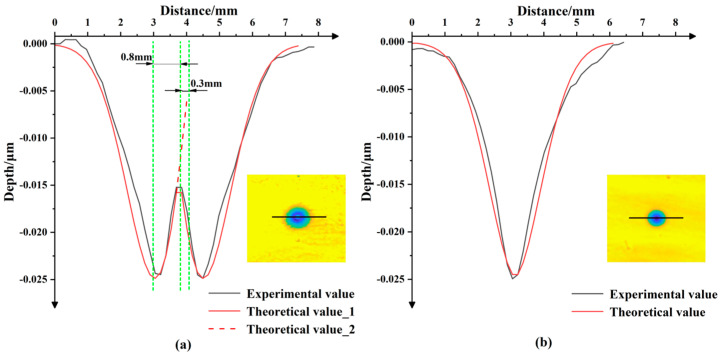
(**a**)The experimental TIF curves (**b**) theoretical TIF curves obtained by rotating the nozzle around different rotation centers.

## Data Availability

The data presented in this study are available upon request from the corresponding author. The data are not publicly available due to the data also forming part of an ongoing study.

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
