# Peer review of "Theoretical Modeling Method for Material Removal Characteristics of Abrasive Water Jet Polishing under Rotating Oblique Incidence"

_micromachines, 2022, doi:10.3390/mi13101690_

Round 1

Reviewer 1 Report

The paper discribes an elaborate modeling approach for a very specific polishing process. Unfortunately it contains a number of shortcomings and therefore cannot be recommended for publication as is. Instead, it requires substantial improvements in various aspects. The following points are only the most protruding examples of deficiencies whithin the manuscript.

Content
- The authors should precisely describe what is meant by the term "TIF"
- The kinematics of the process is not sufficiently described. The reader can only imagine what kind of roation is taking place
- give reasons why the relative velocity v of the preston equation can be replaced by a wall shear stress (line 105)
- what is the purpose of equations (6) and (7)? What is the point of fitting curves that were numerically calculated? Why are pressure and velocity expressed as a function of time and the angular velocity instead of the actual angle?
- for the important comparison between simulation and experimental result, the authors refer to equ. (39) which does not exist!

Formal Aspects
- formula symbols are often used inconsistently
    - upper and lower case symbols are mixed up
    - the latin symbol "v" is used instead of the greek symbol "nu" (line 130)
- please name the original source for the "Preston hypothesis"
- Figures 4, 7, 9 and 11 do not contain a color scale for the depicted quantity
- Fig. 4 seems to contain some markings. However, they are illegible
- the discussion of Fig. 4 (line 199 et seq.) is incomprehensible
- Fig. 9 is followed by Fig. 11
- Chapter 3 contains a number of defective references
- referencing format is not entirely consistent
- the authors seem to cite only two sources from outside the Peoples Republic of China. Does this really reflect the state of the art?

Language
- there is a number of incomprehensible terms within the manuscript. Examples:
    - "cultivation", line 41
    - "high nozzle range", line 75
    - "change law of pressure", line 172
- there is a number of inadequate terms within the manuscript. Examples:
    - "scholars home and abroad", line 56
    - "Li [14] believes...], line 69
- the use of definite and indefinite articles should be improved within the entire manuscript

Reviewer 2 Report

This paper established a theoretical model of the material removal for the abrasive water jet polishing with rotating oblique incidence by analyzing the collision process between the abrasive particle and the workpiece and the material removal by the collision of a single particle. Overall, the work is interesting and worth publication provided that the points below are clarified:

1. For an oblique liquid jet impinging on a solid surface, the pressure distribution at the solid surface should be related to the jet angle. Please explain why the equation (3) does not include the jet angle?

2. Is Pm in equations (3) and (4) the maximum pressure of the fluid at the stagnation point? If so, the density in the equation (4) should be the density of the liquid.

3. Page 8, line 273: “the velocity decays rapidly after the fluid enters the impact zone”. Please explain why “the velocity of the abrasive particles before the collision with the workpiece is um0” can be assumed?

4. The equation (15) is essentially the integral formula for the volume of a spherical cap. It is better to give the volume formula directly.

5. In the experiments, how to determine the position of the stagnation point and how to ensure that the nozzle rotates around the axis where the stagnation point is located?

Round 2

Reviewer 1 Report

The authors provided extensive remarks but only improved parts of the manuscript.

Example:  no additional arguments are given, why the velocity "v can be characterized as the wall shear stress"

Also, the manuscript still contains formal errors. Examples:

- Pressure and velocity are still symbolized by capital letters P and V instead of p and v.

- For Poisson's ratio "µ" ("mu") ist used instead of "n"("nu").
